# Clinical Predictors of Grade Group Upgrading for Radical Prostatectomy Specimens Compared to Those of Preoperative Needle Biopsy Specimens

**DOI:** 10.3390/diagnostics12112760

**Published:** 2022-11-11

**Authors:** Masayuki Tomioka, Chiemi Saigo, Keisuke Kawashima, Natsuko Suzui, Tatsuhiko Miyazaki, Shinichi Takeuchi, Makoto Kawase, Kota Kawase, Daiki Kato, Manabu Takai, Koji Iinuma, Keita Nakane, Tamotsu Takeuchi, Takuya Koie

**Affiliations:** 1Department of Urology, Gifu University Graduate School of Medicine, Gifu 5011194, Japan; 2Department of Pathology and Translational Research, Gifu University Graduate School of Medicine, Gifu 5011194, Japan; 3Department of Pathology, Gifu University Hospital, Gifu 5011194, Japan

**Keywords:** prostate cancer, grade group, upgrade, body mass index (BMI), neutrophil-to-lymphocyte ratio (NLR)

## Abstract

Background: Decision-making and selection of treatment modalities for newly diagnosed prostate cancer (PCa) are often determined by risk stratification using grade group (GG), prostate-specific antigen (PSA), and clinical stage. The discrepancies between needle biopsy (NB) and radical prostatectomy (RP) specimens often occur because of the sampling errors in NB or multifocal features of PCa. Thus, we aimed to estimate the preoperative clinical factors for predicting GG upgrading after robot-assisted RP (RARP). Methods: In this retrospective study, we reviewed the clinical and pathological records of patients who underwent RARP at Gifu University Hospital. We focused on patients with organ-confined PCa who had not received neoadjuvant therapy prior to RARP. The primary endpoint was identified as the predictive factor of GG upgrading for RARP specimens compared to those of NB specimens. Results: Eighty-one patients were included in this study. The enrolled patients were divided into two groups: those who had GG upgrading for RARP specimens (the NB upgrade group) or those who did not have GG upgrading (the no upgrade group). The median age of all patients was 70 years, and the median body mass index (BMI) was 22.9 kg/m^2^. The median neutrophil count was 3720/μL, lymphocyte count was 1543/μL, and neutrophil-to-lymphocyte ratio (NLR) was 2.24. In univariate analysis, BMI, PSA, neutrophil count, and NLR were significantly associated with GG upgrading in RARP specimens compared to NB specimens. BMI and NLR were identified as strong predictive factors for GG upgrading in RARP specimens in the multivariate analysis. Conclusions: Although this study’s small number of enrolled patients was a vital weakness, BMI and NLR might have been significantly correlated with GG upgrading for RP specimens compared with NB specimens. Therefore, BMI and NLR may have potential benefits for newly diagnosed patients with PCa in terms of decision-making and the selection of treatment modalities.

## 1. Introduction

Prostate cancer (PCa) is the most prevalent cancer in men, accounting for approximately 30% of all cancers [1]. Decision-making and the selection of treatment modalities for newly diagnosed PCa are often determined by risk stratification using the Gleason score (GS), prostate-specific antigen (PSA), and clinical stage, according to several guidelines [2,3,4]. Among the three factors, GS may have the greatest impact on predicting pathological stage, tumor aggressiveness, and oncological outcomes, such as biochemical recurrence (BCR) or disease progression [5,6]. The grading group (GG) is frequently used and has been proven to be an acceptable predictor of the biological characteristics of PCa instead of GS [7]. GG was accepted by the World Health Organization for the 2016 edition of Pathology and Genetics, and the GG 1–5 terms were proposed as the new grading system by the 2014 International Society of Urological Pathology (ISUP) consensus conference [8]. Additionally, according to several guidelines, the combination of targeted biopsy (TBx) of suspicious lesions on multiparametric magnetic resonance imaging (MRI) and transrectal ultrasound-guided (TRUS) systematic biopsy (SBx) for the prostate has produced more favorable PCa detection rates than was previously possible [2,3,4,9,10].

However, discrepancies between needle biopsy (NB) specimens and radical prostatectomy (RP) specimens often occur due to sampling errors in NB or the multifocal features of PCa [9,10,11]. Several investigators reported that the discordance in GG between NB and RP specimens was observed at frequencies of 31.0–62.8% [12,13]. Therefore, several clinical factors, including PSA, PSA density (PSAD), maximum foci length of NB specimens, neutrophil count, neutrophil-to-lymphocyte ratio (NLR), and body mass index (BMI), have been identified to predict GG upgrade in RP specimens compared with NB specimens [14,15,16,17,18]. Thus, we aimed to estimate the preoperative clinical factors for predicting GG upgrading after robot-assisted RP (RARP).

## 2. Materials and Methods

### 2.1. Patient Population

This study was approved by the Institutional Review Board of Gifu University (Approval No. 2020-086). As this was a retrospective study, the requirement for informed consent from the patients was waived. Written consent was not required, in accordance with the provisions of the Ethics Committee and Ethics Guidelines in Japan. This is because the results of retrospective and observational studies that use existing materials and other sources have already been published. The details of the study can be found at https://www.med.gifu-u.ac.jp/visitors/disclosure/docs/2020-086.pdf (accessed on 27 September 2022).

In this retrospective study, we reviewed the clinical and pathological records of 230 consecutive patients who underwent RARP at Gifu University Hospital. We focused on patients with organ-confined PCa who had not received neoadjuvant therapy prior to RARP. Patients with lymph node involvement, distant metastases, or clinical stage T4 disease, according to the 2010 American Joint Committee on Cancer Staging Manual, were excluded from the study [19]. Patients who had received finasteride or dutasteride prior to prostate biopsy, those who had undergone transurethral resection of the prostate, and those undergoing active surveillance (AS) were also excluded.

The following preoperative clinicopathological and laboratory parameters were collected: age, height, weight, BMI, serum preoperative PSA level, prostate volume (PV), PSAD, GG of NB specimens and that of RP specimens, neutrophil count, lymphocyte count, NLR, C-reactive protein (CRP), and butyrylcholinesterase (BChE) [20].

### 2.2. Prostate Biopsy Protocol

The prostate biopsy method at our institution has been previously reported [10]. Briefly, TRUS-guided transrectal and transperineal tissue biopsies were performed under spinal anesthesia using an 18G automated biopsy gun. All the patients underwent SBx with 12 cores. TBx was performed for suspicious lesions according to biparametric MRI (bpMRI), ranging from two to four cores depending on lesion size.

### 2.3. Statistical Analysis

The primary endpoint was identified as the predictive factor of GG upgrading for RARP specimens compared with NB. JMP Pro version 16.2.0 (SAS Institute Inc., Cary, NC, USA) was used for data analyses. Clinical covariates were compared using Pearson’s chi-square test. Based on the area under the receiver operating characteristic (ROC) curve, the cutoff values for clinical covariates were defined as the minimum value for (1 − sensitivity)^2^ + (1 − specificity)^2^ [21]. Univariate and multivariate logistic regression analyses were used to identify predictive factors for GG upgrading in RP specimens. All two-sided *p* values <0.05 were considered statistically significant.

## 3. Results

### Patients

Eighty-one patients were enrolled in the study (Figure 1).

The demographic data in the enrolled patients are listed in Table 1. A total of 79 patients (97.5%) underwent prostate biopsy through the perineum and 2 (2.5%) through the rectum. Due to the small number of patients who underwent transrectal prostate biopsy, a comparison between the two groups was not possible. All patients underwent SBx, of which 55 (67.9%) received additional TBx.

The rate of upgraded and downgraded GG in RARP specimens compared with NB specimens (Figure 2).

In the No upgrade group, 35% of the enrolled patients had the same diagnosis of GG in biopsy and surgical specimens. Regarding the GG of biopsy and surgical specimens, 32 patients (39.5%) had the same diagnosis for SBx and 21 (38.2%) for TBx. On bpMRI before prostate biopsy, 19 (34.5%) were diagnosed as PI-RADS 3, 31 (56.4%) as 4, and 5 (9.1%) as 5. Among these cases, 6 (31.6%) of PI-RADS 3, 12 (38.7%) of 4, and 3 (60.0%) of 5 had the same GG between biopsy and surgical specimens. Regarding patients with GG 4 in the biopsy specimens, only two patients were downgraded to GG2 in the surgical specimens. Although only the GG4 component seems to have been detected in the prostate biopsy, it is necessary to accumulate more cases for further investigation.

Figure 3 showed that GG was changed from one in NB specimens (Figure 3A) to three in RARP specimens (Figure 3B).

The enrolled patients were divided into two groups: those who had GG upgrading for RARP specimens compared with NB (upgrade group) and those who did not (no upgrade group). Regarding clinical covariates, there were no significant differences between the groups (Table 2).

In univariate analysis, BMI, PSA, neutrophil count, and NLR were significantly associated with GG upgrading for RARP specimens compared with NB specimens (Table 3). BMI and NLR were identified as strong predictive factors for GG upgrading in RARP specimens in the multivariate analysis (Table 3).

## 4. Discussion

To the best of our knowledge, this is the first study to show that BMI and NLR are significantly associated with GG upgrading in RARP specimens compared with NB specimens.

Patients with PCa have various treatment options, including AS, radiotherapy, focal therapy, and surgery, depending on the risk classification of PCa [2,3,4,13,15,16,22]. For clinical decision-making and the selection of treatment modalities in patients with Pca, it may be of utmost importance to accurately diagnose GG in NB specimens [12,13,14,15]. However, several investigators have reported a discrepancy between the GG of specimens collected via NB and those obtained via RP in patients with Pca [12,13,14,15,22]. There are several possible reasons for GG upgrading. First, TRUS-biopsy of the prostate is the most useful method for detecting Pca; however, pathologic errors or sampling errors contribute to GG mismatches between NB and RP specimens [14]. The most common sampling error occurs when biopsies are taken from different locations of the graded component during RP, resulting in an upgrade or downgrade of Pca [23]. Second, Pca is a well-known multifocal and bilateral disease, with an average of 7.3 foci per prostate [11]. In our previous study, 59.6% of patients with a single NB-positive core had bilateral PCa, and 25.8% of patients were diagnosed with extraprostatic extension or seminal vesicle invasion [11]. Therefore, the location and number of positive NB cores in the prostate are unlikely to reflect the disease severity and grade of malignancy of the entire PCa. Furthermore, there was no significant difference in the GG concordance rate between TBx and SBx in surgical specimens (38.2% vs. 39.5%) in this study. Thus, this may not reflect the pathophysiology of PCa status, even though TBx may detect malignant neoplasms of the prostate with high sensitivity [9,10]. To date, evidence regarding their predictors remains scarce, even though many studies have evaluated the discrepancies in GG between NB and RP specimens [23].

In several studies, age, BMI, preoperative PSA, PV, PSAD, number or percentage of NB-positive cores, maximum foci length, and NLR were found to be significant predictors for GG upgrading in RP specimens compared with NB specimens [14,15,16,22,24]. PCa patients diagnosed with a GG upgrade for RP specimens were also more likely to have extracapsular extension (EPE), seminal vesicle invasion (SVI), positive surgical margins (PSM), and lymphatic invasion at the time of RP, which were significantly associated with BCR [13,22,24]. Conversely, Abedi et al. reported that PSA level was not significantly associated with GG upgrading [23]. Additionally, improvements in NB techniques have significantly reduced the risk of GG upgrading for RP specimens [12,25,26]. Thus, the clinical factors that accurately predict GG upgrading before surgery still seem confounded under the present circumstances.

According to several studies, a higher BMI is statistically associated with an increased risk of biologically aggressive PCa, PCa-specific mortality, and BCR [16,26,27,28,29]. Additionally, a higher BMI is linked with higher risk of aggressive PCa while exercise in unrelated to PCa risk [26]. Lavalette et al. reported that men with a normal BMI at the of age 20 years who became overweight or obese in adulthood had an increased risk of aggressive PCa compared with men who maintained a normal BMI [27]. Furthermore, BMI trajectories resulting in overweight or obesity were more strongly associated with aggressive PCa in none-smokers [27]. The association between obesity and PCa is particularly pertinent due to the large number of men affected by both diseases [27]. The assessment of life-course BMI can help identify men who are at increased risk of PCa and may provide new prevention strategies [27]. A meta-analysis of prospective cohort studies including approximately 7000 PCa-related deaths reported a 15% increase in PCa specific mortality for every 5 kg/m^2^ increase in BMI (relative risk [RR]: 1.15; 95% confidence interval [CI], 1.06–1.25) [28]. Another meta-analysis of case–control studies including almost 1,000 PCa-specific deaths reported a 20% increase in PCa-specific mortality for every 5 kg/m^2^ increase in BMI (RR: 1.20; 95% CI, 0.99–1.46) [28]. The magnitude of RRs estimated using these large populations in different study designs indicates a robust association between obesity and fatal PCa [29]. Obesity is a significant predictor of BCR and has been reported to be associated with invasive PCa, suggesting faster PCa cell proliferation in obese men [29]. Therefore, men with a higher BMI produce less testosterone and are consequently more prone to PCa, which has acquired a less androgen-dependent and more aggressive nature [26]. A higher BMI may contribute to a more favorable microenvironment for cancer development and growth [26]. Excess fat may also promote tumor growth through the secretion of various proinflammatory cytokines [26]. In addition, obese men usually have high levels of insulin and insulin-like growth factor 1, both of which inhibit apoptosis and may promote carcinogenesis [26]. Therefore, BMI may be associated with GG upgrading in RP specimens compared with NB specimens in this study.

In recent years, an increased ratio of peripheral NLR has been recognized as an indicator of poor prognosis in various cancers [30]. Regarding the relationship between PCa and NLR, patients with PCa who were diagnosed with GG upgrading in RP specimens had significantly higher EPE, SVI, and PSM [24]. In logistic regression analysis, patients with increased NLR were 1.68 times more likely to show GG upgrading in RP specimens [24]. Lee et al. reported that PCa patients with high NLR had higher GG of NB specimens, pathological GG, and pathological stage than their counterparts [25]. Multivariate analysis revealed that a high NLR was significantly correlated with adverse pathological outcomes, including a high pathological stage and a high rate of EPE [25]. Furthermore, a high NLR was significantly associated with poor BCR-free survival in Kaplan–Meier analysis and was a significant predictor of BCR after RP in multivariate analysis for PCa patients [25]. Another study showed that the AUC of NLR for BCR was 0.824, with a threshold of 2.62, sensitivity of 71.2%, and specificity of 81.6% [31]. Additionally, patients with a high NLR were significantly associated with higher covariates, including preoperative PSA, GG of NB specimens, pathological T stage, and BCR than those with low NLR [30]. A recent study found that, in patients without systemic or prostate-related inflammation, high NLR, platelet-to-lymphocyte ratio, and eosinophil-to-lymphocyte ratio were significantly associated with staging [17]. These findings suggest that these less expensive and more accessible tests deserve more attention as a potential means of selecting eligible patients for active surveillance (AS) [17]. Van Soest et al. reported that higher NLR is a predictor of aggressive disease and drug resistance in patients with castration-resistant PCa [32]. Furthermore, NLR has been suggested to be strongly correlated with GG upgrade and BCR in low-risk PCa patients [17]. The mechanisms underlying the association between high NLR and poor outcomes in patients with cancer are not well-understood [33]. One mechanism by which NLR may affect prognosis is the association between high NLR and inflammation [33]. Chemokines and inflammatory cytokines can be produced by both tumors and associated host cells, such as neutrophils, and can contribute to malignant progression [34]. An elevated NLR may be associated with increased macrophage infiltration into peritumoral tissues and elevated interleukin 17 levels [35]. Based on the recent study, the association between subpopulations of white blood cells (WBCs) and high GG has been studied, and it was found that the serum monocyte fraction of WBCs was significantly increased in patients with GG ≥ 2 [36]. Özsoy et al. reported that preoperative NLR ≥3 is associated with adverse pathological features, including higher RP GG, SVI, and nodal involvement [18]. Preoperative knowledge of NLR was most helpful in predicting GG upgrade from biopsy to RP, especially for patients who could consider AS [18]. Given the strong prognostic power of biopsy GG in predicting clinical course and treatment efficacy in almost all stages of PCa, knowledge of preoperative NLR would add information to improve clinical decision making [36]. Guner E et al. reported that the presence or absence of findings of chronic prostatitis on prostate biopsy is associated with the upgrading of surgical specimens [37]. This also supports the hypothesis that inflammation is associated with prostate cancer upgrading [37]. In this study, the direct causal relationship between GG upgrading for RP specimens and the NLR remains unclear. However, the inflammatory response of PCa and its surrounding tissues may play a role in GG upgrading in RP specimens compared with that in NB specimens.

Our study had several limitations. First, this was a retrospective study and may have an inherent potential for bias. Second, a relatively low number of patients was enrolled, and the follow-up period was relatively short. Therefore, the association between GG upgrading and oncological outcomes, such as BCR or progression-free survival was not evaluated in this study. Third, patients with locally advanced or metastatic PCa and those who received neoadjuvant therapy before RARP were excluded from this study. Therefore, the relationship between potentially high-grade PCa with aggressive features and GG upgrading, especially with regard to oncological outcomes, remains unclear.

## 5. Conclusions

Although this study’s low number of enrolled patients was a vital weakness, BMI and NLR might have been significantly correlated with GG upgrading for RP specimens compared with NB specimens. Therefore, BMI and NLR may have potential benefits for newly diagnosed patients with PCa in terms of decision-making and the selection of treatment modalities. Further studies involving a larger number of patients with long-term follow-up are warranted.

## Figures and Tables

**Figure 1 diagnostics-12-02760-f001:**
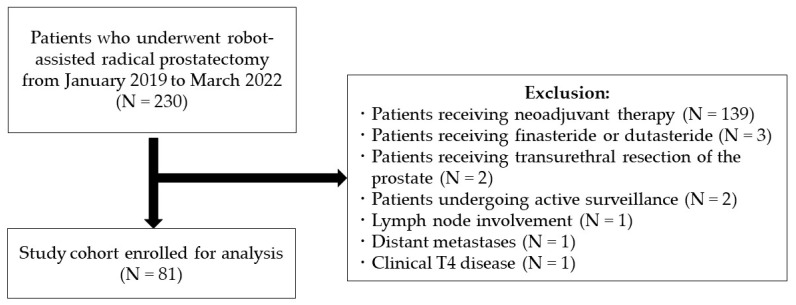
Flow diagram of the study cohort.

**Figure 2 diagnostics-12-02760-f002:**
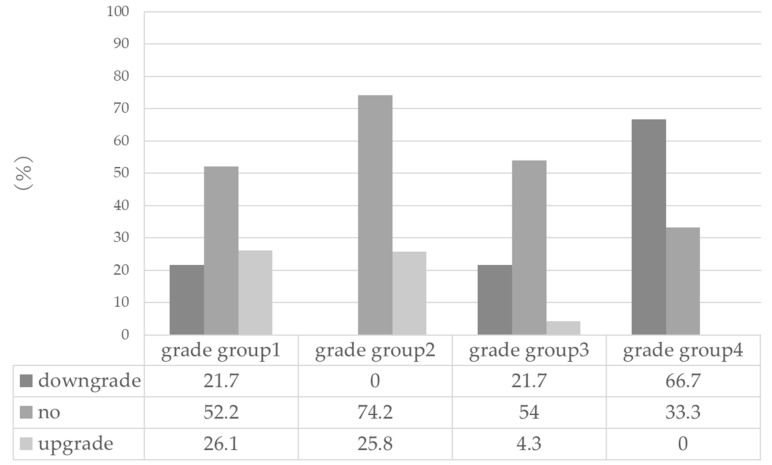
The rate of grade group upgrading and down grading in robot-assisted radical prostatectomy specimens compared with needle biopsy specimens.

**Figure 3 diagnostics-12-02760-f003:**
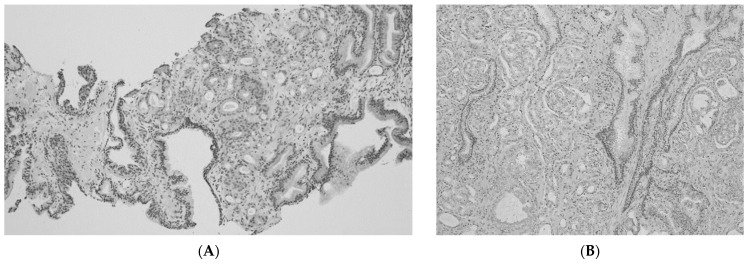
(**A**) Grade group one in the needle biopsy specimen. (**B**) Grade group three in the robot-assisted radical prostatectomy specimens. magnification: ×40.

**Table 1 diagnostics-12-02760-t001:** Demographic data for the enrolled patients.

Covariates	
Number of patients	81
Age (years, median, interquartile range)	70.0 (67.0–73.0)
BMI (kg/m^2^, median, interquartile range)	22.9 (21.4–24.7)
PSA (ng/mL, median, interquartile range)	6.4 (4.9–8.3)
PV (mL, median, interquartile range)	30.0 (21.1–37.8)
PSAD (ng/mL^2^, median, interquartile range)	0.22 (0.15–3.14)
Neutrophil count (/μL, median, interquartile range)	3720 (2665–4640)
Lymphocyte count (/μL, median, interquartile range)	1543 (1187–1992)
NLR (median, interquartile range)	2.24 (1.57–3.14)
CRP (mg/dL, median, interquartile range)	0.06 (0.03–0.13)
BChE (U/L, median, interquartile range)	305 (269–346)

BMI = Body mass index; PSA = Prostate specific antigen; PV = Prostate volume; PSAD = Prostate specific antigen density; NLR = Neutrophil-to-lymphocyte ratio; CRP = C-reactive protein; BChE = Butyrylcholinesterase.

**Table 2 diagnostics-12-02760-t002:** Clinical covariates for two groups.

	Upgrade Group	No Upgrade Group	*p*
Number of patients	27 (33.3%)	54 (66.7%)	
Age (years, median, interquartile range)	71.0 (67.0–71.0)	69.5 (66.8–73.0)	0.716
BMI (kg/m^2^, median, interquartile range)	23.0 (21.2–25.1)	22.9 (21.6–24.7)	0.722
PSA (ng/mL, median, interquartile range)	6.4 (4.7–10.6)	6.4 (5.0–7.9)	0.457
PV (mL, median, interquartile range)	30.0 (20.0–37.7)	30.1 (21.1–37.9)	0.815
PSAD (ng/mL^2^, median, interquartile range)	0.22 (0.14–0.38)	0.22 (0.16–0.29)	0.481
Neutrophil count (/μL, median, interquartile range)	3660 (2840–4838)	3745 (2603–4550)	0.458
Lymphocyte count (/μL, median, interquartile range)	1622 (1191–2059)	1475 (1167–1937)	0.505
NLR (median, interquartile range)	2.24 (1.40–3.29)	2.21 (1.61–3.10)	0.598
CRP (mg/dL, median, interquartile range)	0.06 (0.03–0.11)	0.01 (0.02–0.14)	0.349
BChE (U/L, median, interquartile range)	294 (263–326)	311 (269–348)	0.160

BMI = Body mass index; PSA = Prostate specific antigen; PV = Prostate volume; PSAD = Prostate specific antigen density; NLR = Neutrophil-to-lymphocyte ratio; CRP = C-reactive protein; BChE = Butyrylcholinesterase.

**Table 3 diagnostics-12-02760-t003:** Univariate and multivariate analysis.

	Univariate				Multivariate		
Covariates	Risk Factors	Odds Ratio	95% Confidence Interval	*p*	Odds Ratio	95% Confidence Interval	*p*
Age (years)	>68	1.964	0.710–5.434	0.190			
BMI	>25.6	4.857	1.110–21.255	0.024	6.241	0.991–39.316	0.044
PSA	>10.54	3.430	0.973–12.092	0.047	1.672	0.246–11.350	0.597
PV	>52.5	2.227	0.585–8.487	0.232			
PSAD	>0.37	1.853	0.633–5.424	0.257			
Neutrophil count	>2610	4.375	0.920–20.89	0.049	4.419	0.634–30.800	0.102
Lymphocyte count	>1580	1.818	0.720–4.623	0.210			
NLR	>5.20	5.910	1.065–32.800	0.031	19.473	2.180–173.970	0.004
CRP	>0.10	0.700	0.250–1.961	0.500			
BChE	>325	0.510	0.184–1.408	0.190			

BMI = Body mass index; PSA = Prostate specific antigen; PV = Prostate volume; PSAD = Prostate specific antigen density; NLR = Neutrophil-to-lymphocyte ratio; CRP = C-reactive protein; BChE = Butyrylcholinesterase.

## Data Availability

Data and material are provided in this paper.

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
