# Peer review of "Clinical Predictors of Grade Group Upgrading for Radical Prostatectomy Specimens Compared to Those of Preoperative Needle Biopsy Specimens"

_diagnostics, 2022, doi:10.3390/diagnostics12112760_

Round 1

Reviewer 1 Report

This unique article compares the Gleason Grading Group of prostate biopsy and prostatectomy specimens in the diagnosis of prostate cancer.

There is no confirmation or description of the concordance of GG lesions in prostatectomy specimens and prostate biopsies. Since prostate cancers are multifocal disease, the confirmation on this point is requested.

It is stated that the targeted biopsy was performed with MRI at the time of biopsy. I think an evaluation of whether the results of that biopsy were consistent with the results of the prostatectomy specimen is also necessary to describe the accuracy of the biopsy. Much consideration has been given to BMI and NLR, but what about this consideration?

There were many cases of downstaging in the group of patients who had GG 3 or 4 specimens in the prostatectomy specimens, what was the association?

In the table, the items are centered and difficult to read. I think it would be better to put them on the left side. Also, I think it would be easier to read the descriptions if you used abbreviations. (e.g. Prostate-specific antigen→PSA)

Author Response

November 2, 2022

Dear Editor-in-Chief

the Diagnostics

Dear Editor

Thank you very much for the review of our manuscript titled “Clinical predictors of grade group upgrading for radical prostatectomy specimens compared to those of preoperative needle biopsy specimens.”

We sincerely appreciate all valuable comments and suggestions, which helped us to improve the quality of our manuscript. Our responses to the Reviewers’ comments are described below in a point-to-point manner. Appropriate changes, suggested by the Reviewers, have been introduced to the manuscript (track-changes mode in the red color font). Let me emphasize our full readiness to make any further improvements to the manuscript.

We hope that our manuscript will be acceptable for publication in the Diagnostics.

We look forward to hearing from you.

Yours sincerely,

Takuya Koie

Department of Urology

Gifu University Graduate School of Medicine

1-1 Yanagido, Gifu, Gifu 501-1194, Japan

TEL.: +81-582-30-6338

FAX: +81-582-30-6341

e-mail: goodwin@gifu-u.ac.jp

Responses to the reviewer's comments

We would like to thank the Reviewers for taking the time and effort necessary to review the manuscript. We sincerely appreciate all the valuable comments and suggestions, which helped us to improve the quality of the manuscript.

Response to Reviewer 1

The authors appreciate the reviewer’s comments. The authors’ point-by-point responses to the comments are given below.

This unique article compares the Gleason Grading Group of prostate biopsy and prostatectomy specimens in the diagnosis of prostate cancer.

  1. There is no confirmation or description of the concordance of GG lesions in prostatectomy specimens and prostate biopsies. Since prostate cancers are multifocal disease, the confirmation on this point is requested.

Response:

The authors have added the following sentence on line 121:

In the No upgrade group, 35% of the enrolled patients had the same diagnosis of GG in biopsy and surgical specimens.

  1. It is stated that the targeted biopsy was performed with MRI at the time of biopsy. I think an evaluation of whether the results of that biopsy were consistent with the results of the prostatectomy specimen is also necessary to describe the accuracy of the biopsy. Much consideration has been given to BMI and NLR, but what about this consideration?

Response:

The authors have added the following sentence on line 112:

All patients underwent SBx, of which 55 (67.9%) received additional TBx.

The authors have added the following sentence on line 123:

Regarding the GG of biopsy and surgical specimens, 32 patients (39.5%) had the same diagnosis for SBx and 21 (38.2%) for TBx.

The authors have added the following sentence on line 170:

Furthermore, there was no significant difference in the GG concordance rate between TBx and SBx in surgical specimens (38.2% vs. 39.5%) in this study. Thus, it may not reflect the pathophysiology of PCa status even though TBx may detect malignant neoplasms of the prostate with high sensitivity [9,10].

3.There were many cases of downstaging in the group of patients who had GG 3 or 4 specimens in the prostatectomy specimens, what was the association?

Response:

The authors have added the following sentence on line 127:

Regarding patients with GG 4 in the biopsy specimens, only two patients were downgraded to GG2 in the surgical specimens. Although only the GG4 component seems to have been detected in the prostate biopsy, it is necessary to accumulate more cases for further investigation.

4.In the table, the items are centered and difficult to read. I think it would be better to put them on the left side. Also, I think it would be easier to read the descriptions if you used abbreviations. (e.g. Prostate-specific antigen→PSA)

Response:

The authors have added abbreviations in Tables according to reviewer’s recommendation. However, In the table, each factor is placed in the center because it is prepared in accordance with the submission rules.

Reviewer 2 Report

1. What is the number (%) of patients with transrectal versus transperineal biopsy, were there differences between groups/subgroups (although numbers are likely to be small)

2. I think that in figure 2 ISUP group the 5th row is not necessary because there were no patients with this group, this is just confusing

3. How many patients had an MRI (mpMRI?) and what were the findings? PI-RADS? Any correlation between MRI findings? 

4. Guner E et al. they also studied upgrading in open and robotic RP – I think the authors should comment on their findings and include them in the discussion.

Ekrem Guner 1 , Yavuz Onur Danacioglu 2 , Yusuf Arikan 3 , Kamil Gokhan Seker 4 , Salih Polat 5 , Halil Firat Baytekin 6 , Abdulmuttalip Simsek 7 The presence of chronic inflammation in a positive prostate biopsy is associated with an upgrade in radical prostatectomy 2021 September 30; 93(3):280-284. Arch Ital Urol Androl doi: 10.4081/aiua.2021.3.280.

Author Response

November 2, 2022

Dear Editor-in-Chief

the Diagnostics

Dear Editor

Thank you very much for the review of our manuscript titled “Clinical predictors of grade group upgrading for radical prostatectomy specimens compared to those of preoperative needle biopsy specimens.”

We sincerely appreciate all valuable comments and suggestions, which helped us to improve the quality of our manuscript. Our responses to the Reviewers’ comments are described below in a point-to-point manner. Appropriate changes, suggested by the Reviewers, have been introduced to the manuscript (track-changes mode in the red color font). Let me emphasize our full readiness to make any further improvements to the manuscript.

We hope that our manuscript will be acceptable for publication in the Diagnostics.

We look forward to hearing from you.

Yours sincerely,

Takuya Koie

Department of Urology

Gifu University Graduate School of Medicine

1-1 Yanagido, Gifu, Gifu 501-1194, Japan

TEL.: +81-582-30-6338

FAX: +81-582-30-6341

e-mail: goodwin@gifu-u.ac.jp

Responses to the reviewer's comments

We would like to thank the Reviewers for taking the time and effort necessary to review the manuscript. We sincerely appreciate all the valuable comments and suggestions, which helped us to improve the quality of the manuscript.

Response to Reviewer 2

The authors appreciate the reviewer’s comments. The authors’ point-by-point responses to the comments are given below.

  1. What is the number (%) of patients with transrectal versus transperineal biopsy, were there differences between groups/subgroups (although numbers are likely to be small)

Response:

The authors have added the following sentence on line 109:

A total of 79 patients (97.5%) underwent prostate biopsy through the perineum and 2 (2.5%) through the rectum. Due to the small number of patients who underwent transrectal prostate biopsy, a comparison between the two groups was not possible.

  1. I think that in figure 2 ISUP group the 5th row is not necessary because there were no patients with this group, this is just confusing

Response:

The authors have revised Figure 2 according to the reviewer’s recommendation.

  1. How many patients had an MRI (mpMRI?) and what were the findings? PI-RADS? Any correlation between MRI findings?

Response:

We perform bpMRI instead of mpMRI in all patients before prostate biopsy [10].

On line 93, the authors have already described the following sentence:

TBx was performed for suspicious lesions according to biparametric MRI (bpMRI), ranging from two to four cores depending on lesion size.

The authors have added the following sentence on line 124:

On bpMRI before prostate biopsy, 19 (34.5%) were diagnosed as PI-RADS 3, 31 (56.4%) as 4, and 5 (9.1%) as 5. Among these cases, 6 (31.6%) of PI-RADS 3, 12 (38.7%) of 4, and 3 (60.0%) of 5 had the same GG between biopsy and surgical specimens.

  1. Guner E et al. they also studied upgrading in open and robotic RP – I think the authors should comment on their findings and include them in the discussion.

Response:

The authors have added the following sentence on line 251:

Guner E et al. reported that the presence or absence of findings of chronic prostatitis on prostate biopsy is associated with upgrading of surgical specimens [39]. This also supports the hypothesis that inflammation is associated with prostate cancer upgrading [39].

Round 2

Reviewer 1 Report

The new text has been appropriately revised and added.